# Methanogenesis at High Temperature, High Ionic Strength and Low pH in the Volcanic Area of Dallol, Ethiopia

**DOI:** 10.3390/microorganisms9061231

**Published:** 2021-06-06

**Authors:** Jose L. Sanz, Nuria Rodríguez, Cristina Escudero, Daniel Carrizo, Ricardo Amils, Felipe Gómez

**Affiliations:** 1Departamento de Biología Molecular, Universidad Autónoma de Madrid, Cantoblanco, 28049 Madrid, Spain; 2Centro de Astrobiología (INTA-CSIC) Crtera, Ajalvir km 4 Torrejón de Ardoz, 28850 Madrid, Spain; rodriguezgn@cab.inta-csic.es (N.R.); cescudero@cab.inta-csic.es (C.E.); dcarrizo@cab.inta-csic.es (D.C.); ramils@cbm.uam.es (R.A.); gomezgf@cab.inta-csic.es (F.G.); 3Centro de Biología Molecular Severo Ochoa (CSIC-UAM), Cantoblanco, 28049 Madrid, Spain

**Keywords:** methanogenesis, Dallol, polyextreme environment, hyperthermophiles, hyperacidophiles, extreme halophiles, *Methanohalobium*, *Methanosarcina*, FISH, C-CH_4_ fractionation

## Abstract

The Dallol geothermal area originated as a result of seismic activity and the presence of a shallow underground volcano, both due to the divergence of two tectonic plates. In its ascent, hot water dissolves and drags away the subsurface salts. The temperature of the water that comes out of the chimneys is higher than 100 °C, with a pH close to zero and high mineral concentration. These factors make Dallol a polyextreme environment. So far, nanohaloarchaeas, present in the salts that form the walls of the chimneys, have been the only living beings reported in this extreme environment. Through the use of complementary techniques: culture in microcosms, methane stable isotope signature and hybridization with specific probes, the methanogenic activity in the Dallol area has been assessed. Methane production in microcosms, positive hybridization with the *Methanosarcinales* probe and the δ^13^C_CH4_-values measured, show the existence of extensive methanogenic activity in the hydrogeothermic Dallol system. A methylotrophic pathway, carried out by *Methanohalobium* and *Methanosarcina*-like genera, could be the dominant pathway for methane production in this environment.

## 1. Introduction

The Dallol geothermal area in the Danakil Depression (Afar, Ethiopia), is located in a deep water-free basin subsiding to 120 m below sea level forming a caldera structure. Situated at the northern segment of the Afar triple junction, the area is characterized by an attenuated continental crust—less than 25 km thick—with active tectonic and volcanic activity [1,2]. Dallol (1.5 × 3 km) rises about 50–60 m above the surrounding salt plains. The East African rift is an active continental rifting splitting Africa from Arabia. The first registered activity began 45 Ma with the most notable eruptions observable as the 2000 m high Ethiopian plateau, an episode emplaced in 1 Myr at 30 Ma [3]. This event is contemporaneous with the rifting activity of the southern area of the Red Sea responsible for the seafloor spreading [4]. The final evolution of the seafloor spreading activity originated the 300 km Danakil depression where Dallol active region is located. Currently, geothermal activity in Dallol is in the form of hot brine springs. Salts from the deeper deposits of the system are dissolved and surge quickly on the surface when the water evaporates forming small chimneys and associated pools and ponds around them (Figure 1) [5]. The tips of these chimneys end up breaking due to the pressure of the rising hot water.

The physicochemical conditions in the subsurface are different from those on the surface, leading to the formation of anaerobic zones inside the chimneys. The hydrothermal area of Dallol is a polyextreme environment due to the low pH (as low as −1.5), the high temperature (up to 109 °C) and the high content of dissolved salts (up to 50%) of the spring hydrothermal waters [6,7]. In addition, their chemical gas composition is dominated by CO_2_ (96.8 to 99.4% of the total gas sampled) but also with trace quantities of N_2_, H_2_S, CH_4_ and H_2_ [8]. Salts (meanly chlorides, sulfates, sodium, iron) end up precipitating and forming the colorful mineral deposits found in the area (Figure 1). Its bright colors are derived from the mineral content: mainly hematite, akaganeite, jarosite, halite, gypsum, sylvite and carnallite wurtzite [7]. In addition, vapors of hydrochloric acid that arise from the subsoil due to nearby magmatic activity make this system one of the most extreme environments known on our planet.

Despite its extreme characteristics, lipids biomarkers and stable isotopic analysis of organic carbon, sulfate and total sulfur of Dallol evaporites pointed towards microbial metabolisms, suggesting a microbial community composed mainly of thermophilic bacteria along with microbial consortia of phototrophs [9]. Moreover, the existence of archaeal microorganisms inhabiting Dallol geothermal area was demonstrated by using the following methodologies, FISH, electron microscopy and 16S rRNA massive sequencing [10]. However, this finding has been challenged by other authors who suggest that the detected diversity was the product of contamination and the microorganisms observed by electron microscopy were artefacts due to the unusual geochemistry of the environment [6].

The biological formation of methane, the end-product of the anaerobic digestion of organic matter, is restricted to archaea [11]. The only substrates for methanogenesis are carbon dioxide (CO_2_) plus hydrogen (H_2_), acetate and methyl compounds. Sulfate reducers out-compete with methanogens for CO_2_/H_2_ and acetate due to thermodynamic considerations [12]. In hypersaline environments, where high sulfate concentrations are found, methanogens can overcome the sulfate reducers competition using non-competitive substrates like methanol, methylamines or dimethylsulfide [13].

A sampling campaign developed in 2016 showed evidence of methanogenic activity after trace methane measurements in different collected samples. Although hyperthermophilic hydrogenotrophic methanogenesis is well known [14,15,16] and methanogens have been described in hypersaline environments [17,18], the pHs in the geothermal Dallol area are much lower than the lowest pH for which methanogenic activity has been recorded. 

In this work, based on methane production in microcosms, the stable isotope signature of the released methane and methanogen detection using fluorescence in situ hybridization (FISH) and CAtalized Reporter Deposition-FISH (CARD-FISH), we prove the existence of methanogenic activity in samples collected from the Dallol chimneys and pools around them. 

## 2. Materials and Methods

### 2.1. Sampling Sites and Analysis

Sampling locations correspond to a small chimney and fumaroles field in the Dallol volcanic area (14°14′19″ N; 40°17′38″ E). Colored salt precipitates from the small chimneys were sampled. Samples were taken from the fumaroles, both Blackish White/Grayish Fumaroles (samples FNA1 and FNA2) and Yellow Fumaroles (D5 and D6) and from the pools of diverse size and colors: Green Pool (sample D7), a Small Iron Black Pool (sample D1) and an Iron Dark Lake (sample D8). 

Temperature of the water bodies was measured using a Duo Digital Thermometer IRT Mode 2054, for superficial waters and pools and a HANNA Thermistor HI935012, for measuring inside the fumaroles. Physico-chemical parameters (temperature, redox potential, conductivity and salinity) were measured in situ using a multi-parametric probe, YSI 556 MPS. The probes were calibrated in situ each sampling day. For pH HI54710-11 and HI5002 (HANNA) kits were used. Conductivity and salinity were calibrated with the standard solutions HI7030M and HI7034M.

Samples were analyzed with Inductively Coupled Plasma Mass Spectrometry (ICP-MS), model NexION 2000 (PerkinElmer) at the Centro de Astrobiología. For Total Reflection X-Ray Fluorescence (TXRF) analysis a TXRF S2 PicoFox instrument from Bruker with fix geometry was used. The ionic content was measured by Ion Chromatography, IC-Dionex D-600. Both, IC and TXRF, at the SIDI Service of UAM. Water activity was measured using a Novasina LabMaster-aw neo, equipped with an eVc21 filter to protect the sensor against degradation and chemical contamination.

### 2.2. Microcosms

27 mL serum bottles were filled with 20 mL of mineral medium usually used for methanogenic archaea growth (in mg/L): 327 K_2_HPO_4_^.^3H_2_O; 91 KH_2_PO_4_; 280 NH_4_Cl; 100 MgSO_4_^.^7H_2_O; 400 NaHCO_3_; 10 CaCl_2_^.^2H_2_O; 100 yeast extract, pH 6.5–7.0 and trace elements as described by [19] and then, sealed with rubber septa. O_2_ was displaced by flushing with a mixture of N_2_:CO_2_ 80:20. Bottles were inoculated in situ injecting 1 mL of water with precipitates from the different sampling sites. A complete set of bottles were incubated at 45 °C, the temperature detected in the pools (Table 1). Another sub-set inoculated with water and dissolved salts from the yellow and black fumaroles were incubated at 95 °C, close to the temperature of the emerging water from these structures (Table 1). Incubations were kept for six months.

The CH_4_ content in the head space was analyzed by Gas Chromatography in a Bruker Series Bypass 450GC, instrument equipped with a CP2056 0.6 m × 1/8” Ultimetal Cromsorb GHP 100–120 mesh and a CP81073 0.5 m × 1.8 Ultimetasl Hayesep Q80–100 mesh columns with an FID detector at 250 °C. N_2_ was used as a carrier gas.

### 2.3. Stable Isotopic Analysis

Microcosms with high methane production were selected for stable isotope composition of methane analysis. δ^13^C_CH4_ was measured on the head space of the selected bottles using a PreCon preconcentration and combustion GasBench unit coupled to a Gas Chromatograph interfaced with a MAT 253 Isotope Ratio Mass Spectrometer (PreCon-C-GC-IRMS, Thermo Scientific, Bremen, Germany). The PreCon is a fully automated gas pre-concentrator for CH_4_ analysis: the system carries out sample concentration and purification (removal of H_2_O, CO_2_ and condensable gases), cryofocusing and injection onto a gas chromatograph. The system is composed of a chemical trap after sample introduction (for water removal), then three liquid nitrogen cooling traps. The first liquid nitrogen trap removes (water, CO_2_ and minor hydrocarbons), then the sample (CH_4_) flows into a micro-combustion oven (recovered with NiO and heated at 1000 °C). The carbon dioxide product is transferred to a second liquid nitrogen trap (to pre-concentrate), then a third trap to cryo-focus the CO_2_ (Appendix A). The δ^13^C of the carbon dioxide is determined with a MAT 253 IRMS (Thermo Fisher Scientific, Waltham, Massachusetts, USA) and reported in the standard per mil notation (‰). A certified standard from Indiana University was used (Methane#2). The analytical precision of the δ^13^C values were within ±0.5‰. Detailed description of the analytical method can be found in [20].

### 2.4. Fluorescence In Situ Hybridization (FISH and CARD-FISH)

Aliquots from the different sampling points were fixed in situ, at 4% formaldehyde. For liquid samples, 37% formaldehyde stock was added directly to the sample. For solid samples, a solution of 4% formaldehyde in PBS was added only to cover the sample, thus, minimizing the loss of salinity. Once fixed, samples were filtered and processed as described before [10]. For FISH, filter sections were incubated in the hybridization buffer (0.9 M NaCl, 20 mM Tris HCI (pH 8), 35% formamide, 0.01% SDS) and 50 ng of CY3-labeled MSSH859 (specific for *Methanosarcinales* order) or MEB859 (specific for *Methanobacteriales* order) probes (Biomers) [21] and incubated at 46 °C for 2 h. Subsequently, filter sections were transferred to a pre-warmed (48 °C) washing solution (0.08 M NaCl, 20 mM Tris HCI (pH 8), 5 mM EDTA and 0.01 % SDS) and incubated at 48 °C for 10 min. Filters were then washed with milliQ water and dehydrated with ethanol. 

CARD-FISH was performed as described before [22], with some modifications. Filters were immobilized with agarose at 0.1% and dried at 37 °C. Endogenous peroxidases were inactivated as indicated [23]. Samples were permeabilized with lysozyme (10 mg/mL in 0.05 M EDTA, 0.1 M Tris (pH 8) solution) at 37 °C for 1 h and, subsequently, with achromopeptidase (60 U/mL in 0.01 M NaCl, 0.01 M Tris (pH 8) solution) at 37 °C for 30 min. Filter sections were incubated in 300 µL hybridization buffer (0.9 M NaCl, 20 mM Tris HCl (pH 8), 10% dextran sulfate, 0.02% sodium dodecyl sulphate (SDS), 1% blocking reagent and 35% formamide) and 1 µL of HRP-labeled probe working solution (50 ng/µL) at 46 °C for 2 h. Filter sections were transferred to a pre-warmed water bath as described above for 10 min at 48 °C and then filter sections were placed in 50 mL of PBS for 15 min at RT. Afterwards, the tyramide signal amplification was undertaken by placing the filters in 1 mL amplification buffer (1 × PBS (pH 7.6), 2 M NaCl, 10% dextran sulfate, 0.1% blocking reagent) amended with 0.0015% H_2_O_2_ and 1 µg/mL Alexa Fluor 594-labelled tyramide for 45 min at 46 °C. Then, filter sections were washed with PBS for 10 min at RT and rinsed with milliQ water. In FISH and CARD-FISH experiments, negative controls were carried out in parallel with CY3-labeled NON338 probe [24].

After hybridization protocols, filters were counterstained with DAPI (4′,6-diadimino-2-phenylindole) or Syto9 as manufacturer recommends and mounted on a glass slide using Vectashield (Vector Laboratories,): Citifluor (Citifluor) (1:4). Only hybridization signals showing the corresponding DNA staining signal were considered positive. Samples were imaged using a confocal laser scanning microscope LSM710 (Carl Zeiss, Jena, Germany) equipped with diode (405 nm), argon (458/488/514 nm) and helium and neon (543 and 633 nm) lasers. Images were collected with a 63×/1.4 oil immersion lens. Fiji software was used to process images [25].

## 3. Results

### 3.1. Microcosms

The main physicochemical variables of the different sampling sites are shown in Table 1. After evidence of methanogenic activity was found during the 2016, in 2017, microcosms were prepared in situ by inoculating 1 mL of water with precipitates in 20 mL of sterile and anaerobic standard methanogenic mineral medium and incubated for six months under anaerobic conditions in the laboratory furnaces at 45 and 95 °C. Methane content in the head space was measured by GC. The pHs at the end of the microcosm incubations were more acidic than at the beginning, around 5.5 in most of the cases.

During the incubation at 45 °C, the microcosms inoculated with samples from sides with close temperatures, i.e., Small Iron Black Pool (D1) and Iron Dark Lake (D8), showed a striking increase in methane concentration. In addition, high methanogenic activity was also observed in samples from Yellow Fumaroles (D6), although methane production could be detected in most of the microcosms at this temperature (Table 2).

A remarkable degree of methane production was detected in samples inoculated with water from fumaroles incubated at 95 °C, similar the temperature of the water measured in situ emerging from them (Table 1 and Table 2). In one of the microcosms inoculated with a yellow fumarole sample (D6) the amount of methane released was higher than 2% of the gases in the head space (22,300 ppm). In general, the production of methane was one order of magnitude greater at 95 °C than at 45 °C (Table 2). Thus, it can be assumed that, in the water emerging from the fumaroles, hyperthermophilic methanogenic archaea able to thrive at 95 °C are present. 

### 3.2. Carbon Isotopic Signatures

Six samples (three dilution each) were analyzed for carbon (δ^13^C_CH4_) isotopic composition. Appendix A shows the carbon isotopic values for the analyzed samples, where isotopic values display a range between −37.0 to −32.6 ‰.

### 3.3. Fluorescence In Situ Hybridization

With the aim of confirming the presence of methanogenic archaea in the Dallol geothermal system, FISH was performed on salts and liquid samples. Positive signals were obtained using MSSH859 probe, specific for *Methanosarcinales*, in salt precipitates of the Blackish White/Grayish Fumarole (sample FNA-1), the Yellow Fumarole (sample D6), the ponds formed around the Green Pools (sample D7 salt) and in the Iron Dark Lake (sample D8) (Figure 2A–D). All FISH positive signal were visualized in salt precipitates and no hybridization signals were detected in liquid samples. Because FISH signal intensity depends on the number of ribosomes microorganisms contain, we resorted to CARD-FISH to detect all microorganisms regardless of their ribosomal content. CARD-FISH hybridizations corroborated the data obtained by FISH and members of the *Methanosarcinales* group were exclusively detected in precipitated salts (Figure 2E–H). No positive signals were observed with the MEB859, specific for *Methanobacteriales*) probe. While the total number of microorganisms is at the order of 10^5^ microorganisms/gr in all samples, the number of detected *Methanosarcinales* ranged between 10^3^ and 10^4^ microorganisms per gram of salt.

## 4. Discussion

Temperature is not a determining factor to exclude methanogenic activity in the Dallol polyextreme ecosystem. Hyperthermophilic methanogens using H_2_ to reduce CO_2_, e.g., *Methanothermus* (optimal temperature between 80–88 °C), *Methanopyrus* (optimum 98 °C), or *Methanocaldococcus jannaschii* (optimum 85 °C) are well-known [14]. The only thermophilic acetoclastic methanogenic archaea species described so far, *Methanosaeta thermophila*, has an optimal temperature between 55 and 60 °C [26], which is compatible with the measured temperatures of Iron Dark Lake and Green Pools, although not with the hot water emerging from the fumaroles. High salt concentrations do not exclude either the presence of methanogens. Several genera have been isolated from hyperhalophilic environments. Most halophilic or highly salt-tolerant methane bacteria belong to the *Methanosarcinaceae* family, i.e., *Methanohalobium*, *Methanohalophilus*, *Methanosalsum*. They are all methylotrophic methanogenic types that obtain energy from methyl group-containing compounds [18]. *Methanohalobium* is the only moderately thermophilic and the most halophilic of all methanogens described. The two species described within the genus *Methanosalsum* are alkaliphilic, so their presence should be ruled out in Dallol.

Although neither extreme temperatures nor salinities exclude the presence of methanogenic archaea, the combination of both factors makes it much more restrictive. Concerning metabolism, as far as we know, no hyperthermophilic methanogenic methylotroph has been described up to now. In addition, only one halotolerant hydrogenotrophic methanogen has been described, *Methanocalculus halotolerans* [27]. In addition, no truly halophilic or hyperthermophilic aceticlastic methanogens have yet been isolated.

The main obstacle to explaining methanogenesis in the Dallol ecosystem is the low pH. The bulk pHs of the different samples (Table 1) are much more acidic than the lowest pH reported so far for any methanogenic activity. It can be argued that in situ methanogenesis might be happening in microniches on the sediments of the pools or in the walls of the fumaroles from which they are dragged out by the emerging hot waters. It is known that flocs facilitate the survival of microorganisms developing in alkaline environments, reducing the pH stress [28], or that biofilm structures can have a different pH than the surrounding environment [29], generating microenvironments which would promote the survival of microorganisms in such extreme conditions. The presence of methanogenic archaea in microniches with environmental variables compatible with methanogenesis, but very different from the basic habitat conditions in the ecosystem (pH and redox potential), have been described in an extreme acidic environment [30]. 

The carbon isotopic (δ^13^C) composition of methane has been used to discriminate between biogenic (<−50 ‰) and abiogenic (thermogenic, >−50%) origins [31]. The carbon isotopic composition of methane also can be used to distinguish among the different biogenic origins, e.g., acetoclastic vs. hydrogenotrophic methanogenesis, due to the carbon isotopic fractionation between the source and methane [32]. Even though less is known about the methane (δ^13^C) isotopic composition produced by methylotrophs using non-competitive substrates, previous studies using cultured methanogens suggest a more depleted signature than the two other pathways [33,34,35]. 

The relatively high isotopic values for this biogenic methane (−32–6 to −37.0‰) could be related to methanogens operating at low substrate concentrations: values for total organic carbon in similar samples were less than 0.2% dry weight [9]. These substrate limitations decrease the isotopic fractionation during methanogenesis, resulting in enriched values of the biological methane. Similarly enriched values for biogenic methane (δ^13^C_CH4_ −31 to −40‰) were found on endoevaporitic hypersaline environments in Baja California [36,37]. 

From among the methanogenic hybridization probes used in this work only the probe against *Methanosarcinales* gave positive signals. The family *Methanosarcinaceae* includes two remarkable genera: *Methanosarcina* and *Methanohalobium*. *Methanosarcina* is the most versatile genus of the methanogens. From a metabolic point of view, it is the only one able to perform the three different types of methanogenesis (hydrogenotrophic, acetoclastic and metylotrophic). In addition, even if it is generally assumed that methanogenesis operates only at circumneutral pH, the presence of *Methanosarcina* was reported in the sediments of an extreme acidic river [30], in peat bogs with pH values ranging from 4.2 to 4.8 [38] and was the only active methanogen in an anaerobic reactor fed with methanol and operating at pH 4.2 [39]. For its part, *Methanohalobium* is extremely halophilic (optimum 4.3 M NaCl, with a range of 2.6–5.1 M); moderately thermophilic (optimum 40–55 °C; maximum, 60 °C); growing between pH 6.0 and 8.3 [40]. *Methanohalobium* is strictly methylotrophic; acetate and H_2_/CO_2_ are not utilized. *Methanosarcina* and *Methanohalobium* are spheroid shaped, regular balls or irregular spheroid body respectively, both growing in small aggregates, compatible with what was observed by FISH/CARD-FISH (Figure 2).

Our results strongly suggest that the detected methanogens belong to the *Methanosarcinaceae* family. Positive hybridization with the MSSH859 probe in saline precipitates and lack of hybridization in both, saline and water samples, with the MB1174 probe, point to *Methanohalobium*-like methanogenic archaea. However, the presence of *Methanosarcina*-like archaea, the most acid-tolerant methanogens and the only genus able to use hydrogen within *Methanosarcinaceae* family, cannot be ruled out.

Several methanogenic strains have recently been isolated from methanogenic enrichments cultures from hypersaline lakes. These strains were moderate thermophiles (optimum at 50 °C), extreme halophiles (optimum at 4 M total Na^+^) and they were neutrophiles or obligate alkaliphiles. They utilize methylated compounds as electron acceptors and hydrogen as electron donor, but no just one of these substrates independently, not even acetate. These new methanogens have been affiliated to a new deep lineage of euryarchaeota, the class *Methanonatronarchaeia* [41]. Considering the polyextreme conditions of the studied environment, it should not be ruled out that the methanogens inhabiting the Dallol geothermal area belong to a new phylogenetic group.

## 5. Conclusions

For the first time, a range of complementary evidence, i.e., methane production in microcosms, stable isotope signature and fluorescence in situ hybridization, demonstrate the presence of extensive methanogenic activity in the hydrogeothermic Dallol system. Taking into account that acetate at pH lower than 5 is extremely toxic for methanogens; that H_2_ has been identified in Dallol; and that methylated amines are formed in hypersaline anaerobic environments as degradation products of glycine betaine that serve as an osmotic protector in many halophilic prokaryotes (18), it is reasonable to assume that the methane is produced by hydrogenotrophic and/or methylotrophic pathway.

## Figures and Tables

**Figure 1 microorganisms-09-01231-f001:**
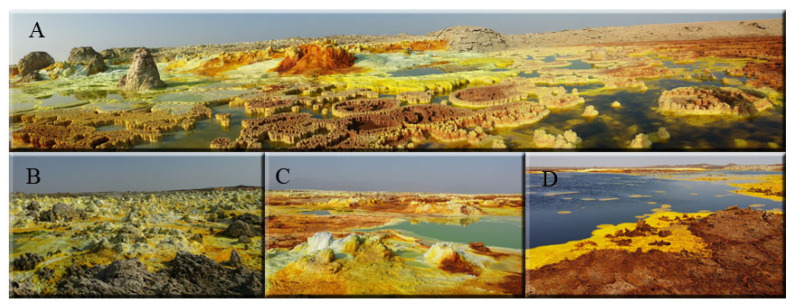
Geothermal Dallol area. (**A**), general view of the area. (**B**,**C**), Yellow and Grayish fumaroles. (**D**), Iron Black Pool.

**Figure 2 microorganisms-09-01231-f002:**
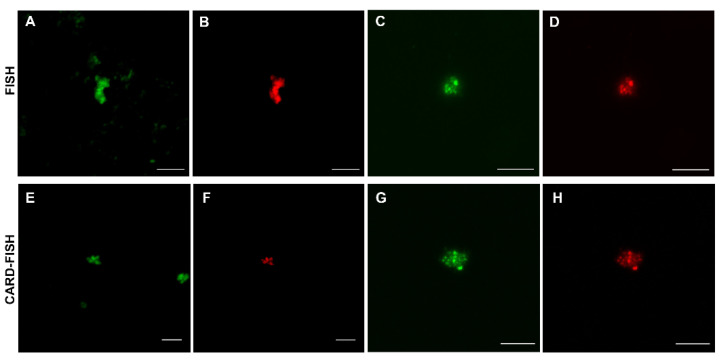
Archaea of the *Methanosarcinales* order inhabiting the Dallol geothermal system detected by FISH (**A**–**D**) and CARD-FISH (**E**–**H**). (**A**,**B**), Yellow Fumarole (D6); (**C**,**D**), ponds formed around the Green Pools (D7); (**E**,**F**), Blackish White/Grayish Fumarole (FNA-1); (**G**,**H**), Iron Dark Lake (D8). In green, DNA stain. In red, MSSH859 probe signal. Scale bars, 5 µm.

**Table 1 microorganisms-09-01231-t001:** Physicochemical characteristics of different sampling points ^a^.

Sampling Point	T (°C)	pH	Conductivity (mS/m^2^)	Water Activity	Cl (g L^−1^)	Na (g L^−1^)	K (g L^−1^)	Mg (g L^−1^)	Ca (g L^−1^)	Fe (g L^−1^)
*Yelow Fumaroles*	98–68 ^b^	0–0.2	188–240	0.7342	192.9 ±4.0	104.7 ±10.1	7.0 ±1.5	3.6 ±0.2	3.2 ±1.5	14.6 ±3.1
*Blackish White/* *Grayish Fumaroles*	101–42 ^b^	0–0.4	n.d.	0.7309	193.8	94.5 ±2.3	6.9 ±1.9	4.0 ±0.4	3.6 ±1.5	14.8 ±1.3
*Green Pools*	37–47	2–2.4	262–270	0.7269	194.1	91.0 ±2.2	9,1 ±4.6	4.9 ±0.8	3.6 ±1.4	17.1 ±2.6
*Iron Dark Lake*	32	0.4	222	0.6970	235	61	28	8.9 ±0.1	1.4 ±0.5	54.6

^a^: For the values of the variables measured in situ (T, pH, salinity and water activity) a range of values measured in different sites with similar characteristics are given. In the case of elements, the given values correspond to the measurements performed with different analytical methods: IC and ICP-MS quantitative and semiquantitative, performed in samples from different sites with similar characteristics. Standard deviations are included. ^b^: The temperature of the water emerging from the fumaroles was close to 100 °C. The water temperature drops after emerging and accumulating in the pools around the fumaroles. Samples were always taken as close as possible from the water emerging from the fumaroles.

**Table 2 microorganisms-09-01231-t002:** Methane production in the microcosms incubated at 45 and 95 °C.

Sample Sites		CH_4_ (ppm)	
		45 °C	95 °C ^1^
Yellow Fumaroles	D5 D6	445 5779	n.d. 22,300
Blackish White/ Grayish Fumaroles	FNA1 FNA2	249 --	7600 11,300
Small Iron Black Pool Iron Dark Lake	D1 D8	8397 3351	
Green Pool	D7	199	

^1^: At 95 °C only inoculum from samples from fumaroles with similar temperature (Table 1) were incubated.

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
