# Peer review of "Methanogenesis at High Temperature, High Ionic Strength and Low pH in the Volcanic Area of Dallol, Ethiopia"

_microorganisms, 2021, doi:10.3390/microorganisms9061231_

Round 1

Reviewer 1 Report

The report is devoted to the study of methanogenic microorganisms inhabiting the Dallol geothermal area. For the first time evidences obtained for methane production in microcosms inoculated by volcano region water taken from polyextreme condition environments (high temperature up to 102 C, low acidic pH up to 0-0.4 and saturating salt concentrations. Authors made an assumption that the methane is produced by hydrogenotrophic and/or methylotrophic pathway. Additionally, they suggested that methane is produced by methanogenic archaea, it might be Methanohalobium-like or Methanosarcina-like archea, and it can not be excluded that a new methanogens inhabiting the Dallol geothermal area belong to a new phylogenetic group.

The authors used a variety of modern experimental methods to achieve their research goal and successfully found archae strains capabable of methane production.. The methodology and analysis of the results are clearly presented and the results obtained are not in doubt.

The following are comments and suggested corrections to a few sentences with the following notations:  [...] - for inclusion and   ]...[ - for deletion:

Introduction:

56: Is it a "minus" before 1.5? If not, then it should be removed here and elsewhere: "... as low as -1.5..."

89:    remove: (17,18]     insert: [17,18]

Results:

219: ]closed[     [close]  temperatures

275:   salt-tolerant  ]methanebacteria[      [methane bacteria]

284:   ]non[   [no] hyperther-284 mophilic methanogenic methylotroph

287: And   ]not[  [no] truly halophilic

300-301:   ... but very different from the ]bulk conditions[   [basic habitat conditions in]    ]of[   the ecosystem   ]that inhabit[  (pH and redox potential) ...

307-308:   ... due [to] the carbon isotopic fractionation between the source and ]the[ methane ...

312: Verify, what does that mean?    

-32-6

324:   From a metabolic point of view [it] is the only one

355-358:  This part should be deleted

Conclusions:

364:  H2 has   ]being[   [been]  identified

Reviewer 2 Report

The authors submitted very interesting article describing methanogenic activity in volcanic area of Dallol. The text is properly arranged, quality of the results presentation is also proper. The presented results are often referenced to the literature what I appreciate. The manuscript brings new knowledge showing that in higly extreme environment (100C, pH 0-2)  bacteria from Methanosarcinaceae family are able to survive what proved they methanogenic activity, converting CO2 and H2 into CH4. I find that results very interesting and I recommend publication.

Author Response

Thank you for your kinds words. We greatly appreciate the time devoted to our manuscript